# Identification of a peptide motif that potently inhibits two functionally distinct subunits of Shiga toxin

Miho Watanabe-Takahashi [1,6], Masakazu Tamada[1,6], Miki Senda [2,6], Masahiro Hibino[1], Eiko Shimizu[1], Akiko Okuta[3,7], Atsuo Miyazawa[4], Toshiya Senda[2,5 ✉] & Kiyotaka Nishikawa [1 ✉]

Shiga toxin (Stx) is a major virulence factor of enterohemorrhagic *Escherichia coli*, which causes fatal systemic complications. Here, we identified a tetravalent peptide that inhibited Stx by targeting its receptor-binding, B-subunit pentamer through a multivalent interaction. A monomeric peptide with the same motif, however, did not bind to the B-subunit pentamer. Instead, the monomer inhibited cytotoxicity with remarkable potency by binding to the catalytic A-subunit. An X-ray crystal structure analysis to 1.6 Å resolution revealed that the monomeric peptide fully occupied the catalytic cavity, interacting with Glu167 and Arg170, both of which are essential for catalytic activity. Thus, the peptide motif demonstrated potent inhibition of two functionally distinct subunits of Stx.

[1] Department of Molecular Life Sciences, Graduate School of Life and Medical Sciences, Doshisha University, Kyoto, Japan. [2] Structural Biology Research Center, Institute of Materials Structure Science, High Energy Accelerator Research Organization (KEK), Ibaraki, Japan. [3] Cellular and Structural Physiology Institute, Nagoya University, Nagoya, Japan. [4] Graduate School of Life Science, University of Hyogo, Hyogo, Japan. [5] Department of Materials Structure Science, School of High Energy Accelerator Science, The Graduate University of Advanced Studies (Soken-dai), Ibaraki, Japan. [6]These authors contributed equally: Miho Watanabe-Takahashi, Masakazu Tamada, Miki Senda. [7]Deceased: Akiko Okuta. ✉email: toshiya.senda@kek.jp; knishika@mail.doshisha.ac.jp

Shiga toxin (Stx) is a major virulence factor of enterohemorrhagic *Escherichia coli*, which causes bloody diarrhea, hemorrhagic colitis, and sometimes fatal systemic complications, such as acute encephalopathy and hemolytic-uremic syndrome[1–6]. Stx is classified into two subgroups, Stx1 and Stx2, each with subgroups of various closely related subtypes[7]. Stx1a and Stx2a are two major subtypes, and Stx2a, in particular, which is highly toxic when injected into mice or primates[8,9], has been linked to systemic complications in humans[6]. Stx consists of a catalytic A-subunit and a B-subunit pentamer. The catalytic A-subunit is an RNA *N*-glycosidase that cleaves a specific adenine from 28S ribosomal RNA to inhibit eukaryotic protein synthesis[10]. It is classified as a ribosome-inactivating protein as is ricin, a phytotoxin isolated from the seeds of the castor plant, *Ricinus communis*[11,12]. The structures of the catalytic regions of Stx and ricin are highly similar[13,14], allowing high-throughput screening of chemical libraries for small compounds that inhibit both toxins by targeting the "adenine-specificity" pocket[15,16]. In contrast, a peptide inhibitor against the catalytic site, which is expected to occupy a wider region of the cavity, has not been identified.

The B-subunit pentamer is responsible for high-affinity binding of Stx to the functional cell surface receptor, Gb3 (Galα[1–4]-Galβ[1–4]-Glcβ-ceramide)[4,17]. Each B-subunit has three distinct binding sites (sites 1, 2, and 3) for the trisaccharide moiety of Gb3[18,19]. This multivalent interaction leads to a million-fold increase in binding affinity, which is referred to as the "clustering effect." Based on this, we previously developed a library of tetravalent peptides designed to exhibit this clustering effect. Screening of the library led to the identification of tetravalent peptides that bind to the B-subunit pentamer and inhibit its toxicity in vitro and in vivo[20–22]. One of these peptides, MMA-tet (containing the following motif, Met-Met-Ala-Arg-Arg-Arg-Arg), inhibits the major Stx family members (e.g., Stx1a and Stx2a), as well as other highly virulent Stx2 subtypes, including Stx2d and Stx2c[23].

In this study, we identified a tetravalent peptide with greater potency than MMA-tet that inhibited both Stx1a and Stx2a by targeting the B-subunit pentamer through a multivalent interaction. The tetravalent peptide has a synthetic amino acid, β−Ala, in each functional motif. Unexpectedly, a monomeric peptide with the same motif, which was unable to exert the clustering effect, also inhibited the cytotoxicity of Stx1a and Stx2a. Crystallographic analysis showed that the monomeric peptide fully occupied the catalytic cavity of the A-subunit.

## Results

### Identification of a tetravalent peptide that potently inhibits Stx by targeting the B-subunit pentamer.

Previously, we identified a tetravalent peptide MMA-tet (containing the following motif, Met-Met-Ala-Arg-Arg-Arg-Arg) that efficiently inhibits Stx1a

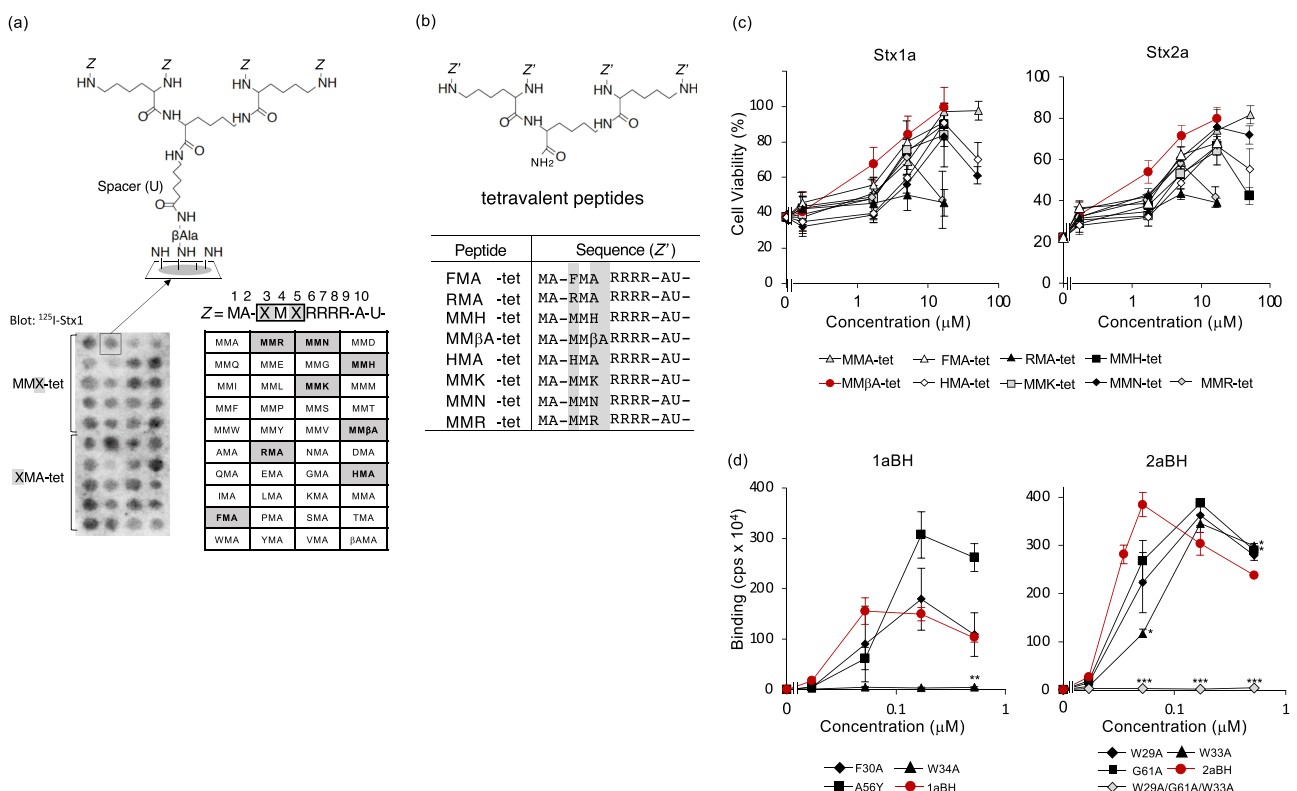

**Fig. 1 Identification of a tetravalent peptide that potently inhibits Stx by targeting the B-subunit pentamer. a** Structure of the tetravalent peptides synthesized on a cellulose membrane is shown. The density of tetravalent peptides on the membrane and the spacer length was optimized for the screening using Stx as follows. The density was set to 100% by using Fmoc-β-Ala-OH without Boc-β-Ala-OH for the first peptide synthesis cycle, and the number of aminohexanoic acid residues (U; spacer length) was set to one (upper panel). Met or Ala at position three or five of MMA-tet, respectively, was replaced by the indicated amino acid (βA; beta-Ala). The tetravalent peptides synthesized on a membrane were blotted with [125]I-Stx1a (left panel). The screening was performed three times (Supplementary Fig. 1). Eight Stx1a-binding motifs (shaded in the right panel) were identified. **b** Structure of the tetravalent peptide and the identified motifs are shown. **c** MMβA-tet most efficiently inhibited the cytotoxicity of Stx1a and Stx2a. Vero cells were treated with Stx for 72 h in the presence of each peptide. Data are presented as a percentage of the control value (mean ± standard error (SE), $n = 3$). **d** AlphaScreen assay to examine the binding between MMβA-tet and the B-subunit or its mutant. Data are presented as signal intensity (mean ± SE, $n = 3$). ***$P < 0.001$, **$P < 0.01$, *$P < 0.05$ (compared with wild type by Dunnett's test).

and Stx2a by screening a library of tetravalent peptides for high-affinity binding to the B-subunit pentamer[20–22]. The second Met and the Arg-cluster of MMA-tet are essential for high-affinity binding to the pentamer[22]. Each Arg in the cluster interacts with a specific acidic amino acid present in the receptor-binding region of the B-subunit pentamer[20]. Changes in the N-terminal region of MMA-tet to either AAA-tet (containing the following motif, Ala-Ala-Ala-Arg-Arg-Arg-Arg) or MAA-tet (containing the following motif, Met-Ala-Ala-Arg-Arg-Arg-Arg) reduced inhibition of the cytotoxicity of Stx1a and Stx2a compared with MMA-tet, indicating that this region plays an essential role in the inhibitory activity of MMA-tet[22]. To find peptides with stronger inhibitory activity than MMA-tet, we screened a series of tetravalent peptides with randomized amino acids synthesized on a cellulose membrane (Fig. 1a). The density of tetravalent peptides on the membrane and the spacer length was optimized for the screening using Stx as shown previously[23,24]. The membrane was blotted with $^{125}$I-Stx1a, and the radioactivity bound to each peptide spot was quantified and analyzed. We identified eight peptides that bound to the pentamer more strongly than MMA-tet (Fig. 1a, Supplementary Fig. 1) and synthesized these tetravalent peptides, FMA-tet, RMA-tet, MMH-tet, MMβA-tet, HMA-tet, MMK-tet, MMN-tet, and MMR-tet (Fig. 1b). Among these peptides, MMβA-tet (containing the following motif, Met-Met-βAla-Arg-Arg-Arg-Arg), which has a synthetic amino acid βAla in its motif, better inhibited the cytotoxicity of Stx1a and Stx2a, with relative IC50 values (concentration that restored viability to 50% of the cells that were killed when no peptide was added) of 1.9 μM and 2.7 μM, respectively (Fig. 1c).

The binding of MMβA-tet to the Stx1a or Stx2a B-subunit pentamer or its receptor-binding site mutants was examined with the AlphaScreen assay. The binding of MMβA-tet to the Stx1a B-subunit pentamer was not greatly affected by the mutations at site 1 (F30A) or 2 (A56Y) but was completely abolished by the

mutation at site 3 (W34A) (Fig. 1d). In contrast, the binding of MMβA-tet to the Stx2a B-subunit pentamer was somewhat affected by the mutations at site 1 (W29A), site 2 (G61A), or site 3 (W33A), and was completely abolished by triple mutations at sites 1, 2, and 3 (W29A/G61A/W33A) (Fig. 1d). These results indicate that MMβA-tet bound to the B-subunit pentamers of Stx1a and Stx2a through site 3 and sites 1−3, respectively.

**MMβA-mono binds to the catalytic A-subunit to inhibit the cytotoxicity.** Unexpectedly, a monomeric peptide with the same motif, Met-Ala-Met-Met-βAla-Arg-Arg-Arg-Arg-Ala (referred to as MMβA-mono), also inhibited the cytotoxicity of Stx1a and Stx2a with greater potency against Stx2a (relative IC50 = 59 and 9.6 μM, respectively) (Fig. 2a). Since a monomeric peptide cannot exert the clustering effect, binding of MMβA-mono to the A-subunit or the B-subunit pentamer was examined using the AlphaScreen assay. MMβA-mono bound almost exclusively to the Stx A-subunits (apparent Kd for Stx1a or Stx2a A-subunit = 0.032 or 0.05 μM, respectively), not to the B-subunit pentamers, whereas MMβA-tet bound only to the B-subunit pentamers (apparent Kd for Stx1a or Stx2a B-subunit pentamer = 0.03 or 0.07 μM, respectively) (Fig. 2b). In an enzyme-linked immunosorbent assay (ELISA), MMβA-tet, but not MMβA-mono, bound exclusively to the B-subunit pentamers (Fig. 2c).

The binding site of MMβA-mono to the A-subunit was determined by X-ray crystallography using a co-crystal of MMβA-mono and the Stx2a holotoxin, which is a subtype that is most closely associated with the severity of the infection. The co-crystal was obtained by the soaking method. Data collection and refinement statistics are summarized in Table 1. The electron density for MMβA-mono is shown in Supplementary Fig. 2. Diffraction to 1.6 Å-resolution showed that MMβA-mono was bound tightly to the catalytic A-subunit through electrostatic interactions of the Arg-cluster with Asp94 and Glu167 of the

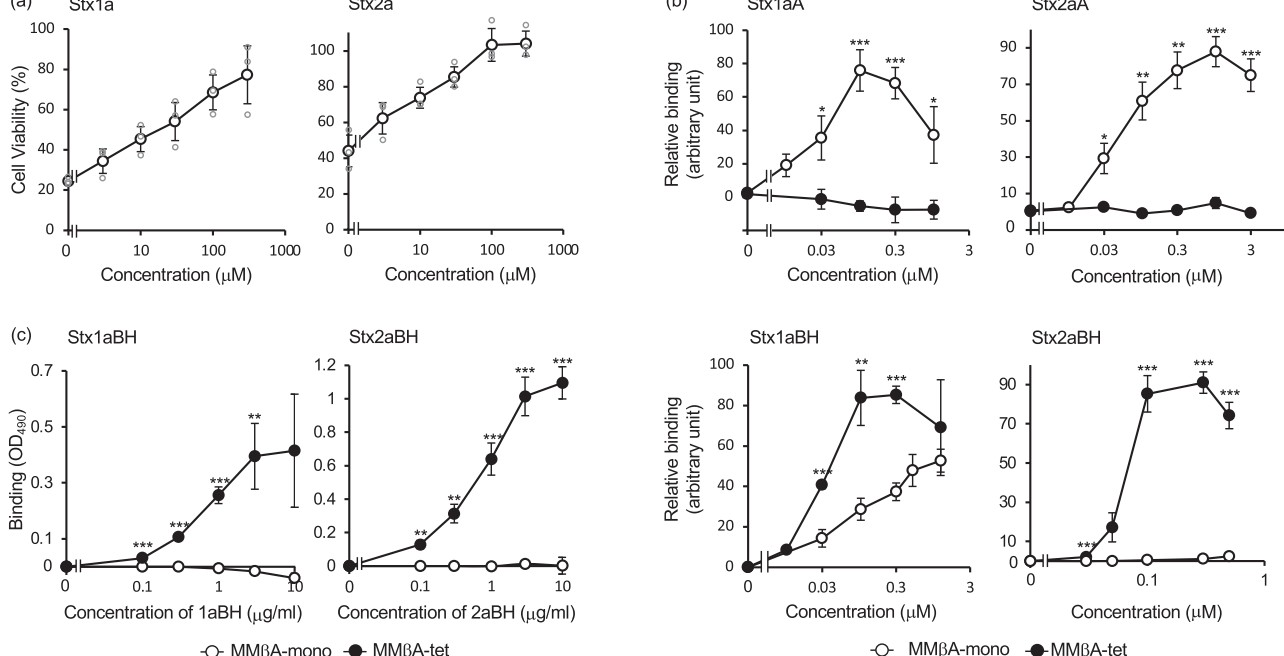

**Fig. 2 MMβA-mono binds to the catalytic A-subunit, but not to the B-subunit pentamer, to inhibit Stx. a** MMβA-mono efficiently inhibited the cytotoxicity of Stx1a and Stx2a. Data are presented as a percentage of the control value (mean ± SE, n = 3). **b** AlphaScreen assay to examine the binding between MMβA-tet/MMβA-mono and the A-subunit (upper panels) or the B-subunit pentamer (lower panels). Data are presented as a percentage of the maximum binding value (mean ± SE, n = 7). **c** Enzyme-linked immunoassay to examine the binding between MMβA-tet/MMβA-mono and the B-subunit pentamer. Data are presented as an Optical Density at 490 nm (OD$_{490}$) (mean ± SE, n = 3). ***P < 0.001, **P < 0.01, *P < 0.05 by Student's t test.

**Table 1 Data collection and refinement statistics.**

|  | Apo | MMA-βAla (MMβA-mono) |
|---|---|---|
| **Data collection** |  |  |
| Space group | $P6_1$ | $P6_1$ |
| Cell dimensions |  |  |
| $a, b, c$ (Å) | 146.7, 146.7, 60.9 | 146.5, 146.5, 60.2 |
| $\alpha, \beta, \gamma$ (°) | 90, 90, 120 | 90, 90, 120 |
| Resolution (Å) | 48.02–1.80 | 73.25–1.60 |
|  | (1.90–1.80) | (1.69–1.60) |
| $R_{merge}$ | 0.104 (0.499) | 0.058 (0.429) |
| I/σI | 25.09 (6.62) | 21.48 (6.37) |
| Completeness (%) | 100.00 (100.00) | 100.00 (100.00) |
| Redundancy | 21.2 (21.5) | 9.8 (10.0) |
| **Refinement** |  |  |
| Resolution (Å) | 48.02–1.80 | 73.25–1.60 |
| No. of reflections | 69378 | 97145 |
| $R_{free}/R_{work}$ | 0.1903/0.1627 | 0.1972/0.1745 |
| No. of atoms |  |  |
| Protein | 4939 | 4928 |
| Peptide | – | 50 |
| PPS | 52 | 52 |
| Water | 465 | 394 |
| B-factors |  |  |
| Protein (Å$^2$) | 16.2 | 20.7 |
| Peptide (Å$^2$) | – | 25.4 |
| PPS (Å) | 23.1 | 28.7 |
| Water (Å) | 24.5 | 29.3 |
| r.m.s deviations |  |  |
| Bond lengths (Å) | 0.007 | 0.006 |
| Bond angles (°) | 0.771 | 0.770 |
| Ramachandran plot |  |  |
| Favored/allowed/outliers | 98.86/1.14/0.00 | 98.87/1.13/0.00 |
| PDB code | 7D6Q | 7D6R |

Values in parentheses are for highest-resolution shell. Each data set was collected from one crystal.

A-subunit and with Asp70 of the B-subunit, residues that are all present in the catalytic cavity (Fig. 3a, b, and Table 2). The replacement of each Arg of MMβA-mono with Ala resulted in a marked reduction in A-subunit binding. Replacement of both Arg residues with Ala at positions 8 and 9 abolished binding, indicating that these two Args are synergistically involved in the binding (Fig. 3c). The C-terminal Ala of MMβA-mono is present in the bottom of the catalytic pocket of the A-subunit; therefore, MMβA-tet cannot bind to the A-subunit because each C-terminal Ala of the motif (Met-Ala-Met-Met-βAla-Arg-Arg-Arg-Arg-**Ala**-) is connected to the core structure through a spacer (see Fig. 1b).

MMβA-mono occupied a wide region of the cavity, including Tyr77, Val78, Asp94, Ser112, Tyr114, Thr115, Glu167, Thr199, and Asp70. In contrast, previously developed Stx2 inhibitory compounds targeting the A-subunit interact with Val78, Ser112, and Arg170 through hydrogen bonds (i.e., adenine)[13], as do inhibitors against ricin, which has a catalytic region that is very similar to the A-subunit[15] (Supplementary Fig. 3). Also, Glu167 and Arg170, both of which are known to be essential for catalytic activity[25,26], were found to interact with Arg at position 8 of MMβA-mono. Inhibition of the N-glycosidase activity of Stx2a holotoxin by MMβA-mono was demonstrated using an in vitro translation assay in which newly synthesized luciferase is measured in a reticulocyte lysate system (Supplementary Fig. 4). Structural superimposition of the Stx2a A-subunit or the Stx1a A-subunit with MMβA-mono resulted in highly similar interactions (Supplementary Fig. 5), suggesting a similar protective effect of MMβA-mono against the cytotoxicity of Stx1a.

## Discussion

Here, we identified MMβA-tet as a potent inhibitor against both Stx1a and Stx2a with greater potency than MMA-tet by screening a series of tetravalent peptides with randomized amino acids synthesized on a cellulose membrane. Similar compounds or peptides that function by targeting the B-subunit pentamer through a multivalent interaction cannot be obtained by other techniques, such as screening of small-molecule libraries or phage display libraries, because these molecules cannot exert the clustering effect. Similarly, MMβA-mono, which has the same functional motif as that of MMβA-tet, but cannot form the multivalent interaction, showed no binding activity to the B-subunit pentamers (Fig. 2b, c), emphasizing the importance of screening possible inhibitors based on the clustering effect.

MMβA-mono bound directly to the catalytic A-subunit, but not to the B-subunit pentamers, to inhibit the cytotoxicity of Stxs. The apparent Kd value of MMβA-mono for the A-subunit of Stx1a or Stx2a (0.032 or 0.05 μM, respectively) is similar to that of MMβA-tet for the binding to the B-subunit pentamer of Stx1a or Stx2a (0.03 or 0.07 μM, respectively), indicating that the binding affinity of MMβA-mono to the A-subunit is similar to that of MMβA-tet to the B-subunit pentamer. In contrast, in the cytotoxicity assay, the relative IC50 value of MMβA-mono for Stx1a or Stx2a (59.2 or 9.6 μM, respectively) was higher than that of MMβA-tet (1.9 or 2.7 μM, respectively), indicating that the inhibitory effect of MMβA-mono was lower than that of MMβA-tet. One possible reason is that the low cell permeability of MMβA-mono might substantially affect inhibition because the target of the A-subunit is 28S ribosomal RNA in the cytosol.

An X-ray crystal structure revealed that MMβA-mono occupied a wide region of the catalytic cavity. Most of the small-molecule inhibitors for which there are co-crystal structures with Stx or ricin occupy the "adenine-specificity" pocket. This pocket recognizes a universally conserved adenosine residue (A$^{4324}$) of 28S rRNA through the interaction with Tyr77, Val78, Tyr114, and Arg170 for Stx2, or Tyr80, Val81, Tyr123, and Arg180 for ricin[13,27]. For each toxin, the two Tyr residues interact with the hydrophobic ring of the adenosine. In this study, we found that all these residues (Tyr77, Val78, Tyr114, and Arg170) are involved in the binding of MMβA-mono to the catalytic pocket (Fig. 3b and Table 2). MMβA-mono also interacted with Asp94 and Glu167 of the A-subunit cavity and with Asp70 of the B-subunit that is adjacent to the cavity. The structural information that defines the unique binding characteristics of MMβA-mono to Stx will be important in determining pharmacophores for the design of more precise and effective small-molecule inhibitors against Stxs. In particular, Arg6 of MMβA-mono, which interacts with Asp94 located in the gate area of the pocket, contributes to the binding to the same extent as Arg8 (Fig. 3c), which interacts with Glu167 and Arg170 that are essential for the catalytic activity, suggesting the importance of Asp94 as a new target for drug development.

It is likely that the N-terminal Met-Met-βAla sequence of MMβA-mono is not essential for the binding of MMβA-mono with the A-subunit, because a series of shorter peptides, such as Ala-Arg-Arg-Arg-Arg-Ala, Arg-Arg-Arg-Arg-Ala, or Arg-Arg-Arg-Arg, also efficiently inhibited the binding between MMβA-mono and the A-subunit (unpublished data). MMβA-mono can function as a binding unit for both the A-subunit and the B-subunit pentamer because the binding region of each subunit is enriched with acidic amino acids that can form stable electrostatic interactions; Asp94, Glu167 of the A-subunit, and Asp70 of the B-subunit are in the catalytic pocket of the A-subunit, and Asp16 and Asp17 of the B-subunit are in the receptor-binding regions, site 1 and site 3, respectively. This is the first report of a peptide motif that can inhibit two functionally distinct subunits of a

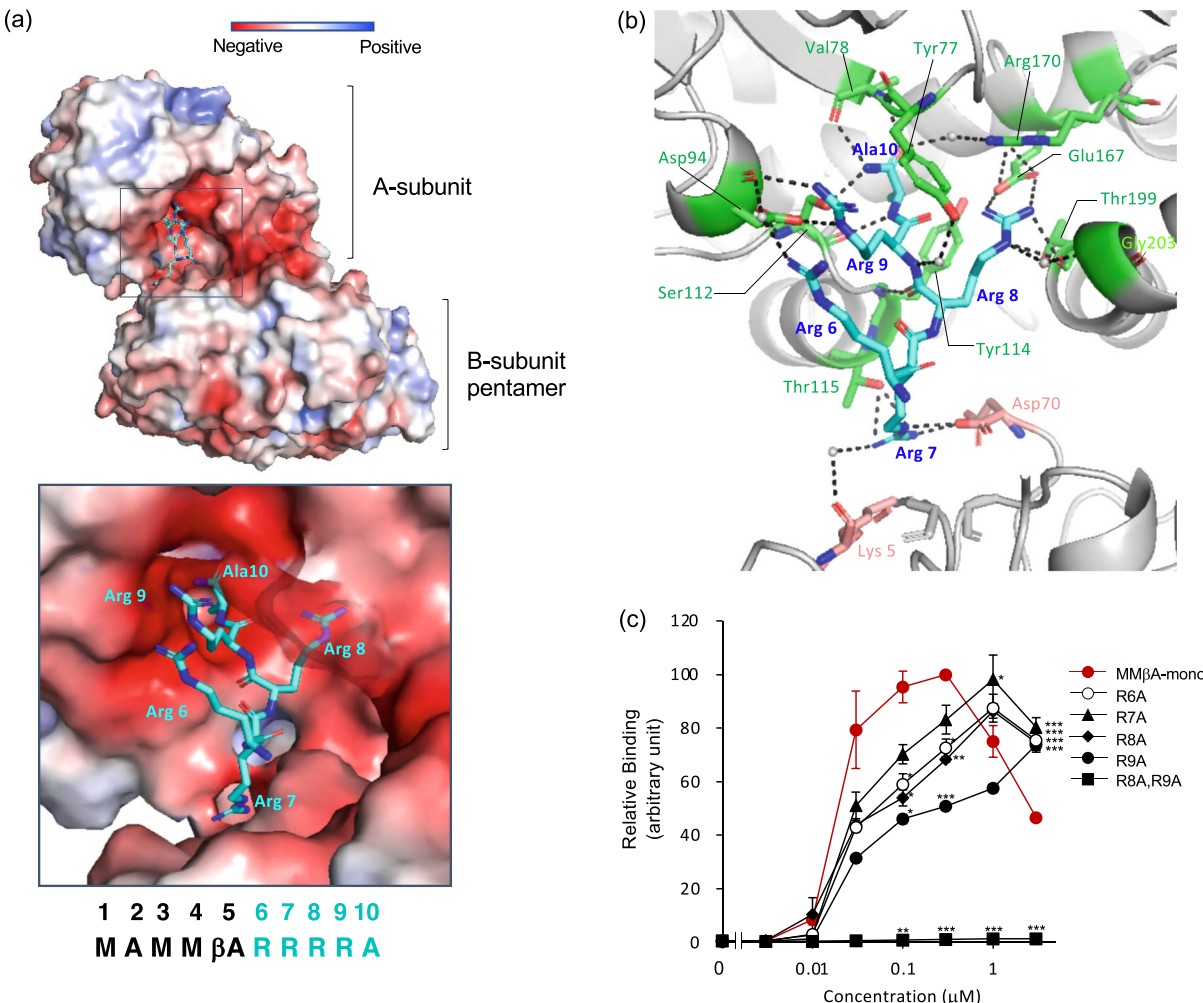

**Fig. 3 Structural analysis of the interaction between MMβA-mono and the Stx2a A-subunit. a** Overall structure of Stx2a holotoxin in complex with MMβA-mono. Stx2a is shown as a charge distribution surface model. The surface of the Stx2a A-subunit is colored by charge (blue, positive; red, negative). MMβA-mono is shown as a stick model. A close-up view shows the catalytic cavity of the Stx2a A-subunit, which is where MMβA-mono binds. **b** Structural view of the binding between the A-subunit and MMβA-mono. Interacting residues are shown as stick models, hydrogen bonds are shown as broken lines, and water molecules are shown as spheres. **c** AlphaScreen assay to examine the binding between the Stx2a A-subunit and MMβA-mono with amino acid substitution(s). Data are presented as a percentage of the maximum binding value (mean ± SE, $n = 3$). ***$P < 0.001$, **$P < 0.01$, *$P < 0.05$ (compared with MMβA-mono by Dunnett's test).

single bioactive molecule, depending on the organization of the peptide structure.

## Methods

**Materials**. Recombinant Stx1a, Stx2a, the histidine-tagged Stx1a B-subunit pentamer (1aBH), 1aBHs containing amino acid substitutions (1aBH-F30A, 1aBH-A56Y, and 1aBH-W34A), the histidine-tagged Stx2a B-subunit pentamer (2aBH), and 2aBHs containing amino-acid substitutions (2aBH-W29A, 2aBH-G61A, 2aBH-W33A, and 2aBH-W29A/G61A/W33A) were prepared as described previously[28]. The Stx1a A-subunit or the Stx2a A-subunit was prepared as follows: purified Stx1a or Stx2a was incubated in a dissociation solution (6 M urea, 0.1 M NaCl, 0.1 M propionic acid pH 4), and each dissociated subunit was separated by gel filtration column chromatography (Sephacryl S-200; GE Healthcare Life Sciences, IL, USA). The fractions containing the Stx A-subunit were dialyzed against 50 mM Tris-HCl (pH 7.4). TentaGel cMAP(4) branched amide resin was used for the synthesis of tetravalent peptides (Rapp Polymere GmbH, Tübingen, Germany), and TentaGel amide resin was used for the synthesis of monomer peptides (Intavis Bioanalytical Instruments AG, Cologne, Germany).

**Screening of tetravalent peptides synthesized on a cellulose membrane**. Tetravalent peptides were synthesized on a cellulose membrane by using a ResPep SL SPOT synthesizer (Intavis Bioanalytical Instruments AG), as described previously[23]. Fmoc-βAla-OH (Watanabe Chemical Industries, Japan) was used in the first cycle, followed by aminohexanoic acid as a spacer. Fmoc-Lys (Fmoc)-OH

(Watanabe Chemical Industries) was used in the next two cycles to create four branches in the peptide chain for subsequent motif synthesis. The Met-Ala sequence and the Ala at the carboxyl terminus of the peptides, respectively, were included based on the structure of MMA-tet. The membrane was blotted with [125]I-labeled Stx1a ([125]I-Stx1a) (1 μg/ml), and the radioactivity of each bound peptide spot was quantified as a pixel value using a BAS-2500 bio-imaging (GE Healthcare Life Sciences). The pixel value was used to evaluate the binding to Stx1a (Supplementary Fig. 1).

**Peptides**. Tetravalent and monomer peptides were synthesized using N-α-Fmoc-protected amino acids and standard BOP/HOB coupling chemistry[20]. The terminal amino groups of the tetravalent peptides or monomer peptides were biotinylated with biotin (Sigma-Aldrich, MO, USA) and 1-(bis[dimethylamino]methylene)-1H-benzotriazolium 3-oxide hexafluorophosphate (Peptide Institute Inc., Japan) in the last cycle of peptide synthesis. The synthesized peptides were validated by mass spectrometry analysis using the AutoflexII TOF/TOF system (Bruker Corp., MA, USA).

**Cytotoxicity assay**. Subconfluent Vero cells were cultured in a 96-well plate in Dulbecco's modified Eagle's medium supplemented with 10% fetal calf serum, 100 units/ml penicillin, 100 μg/ml streptomycin, and 0.25 μg/ml amphotericin B, after which they were treated with Stx1a or Stx2a (3 pg/ml) in the absence or presence of a given peptide or MMβA-mono for 72 h at 37 °C. The relative number of living cells was determined using a cell counting kit (Nacalai Tesque, Japan), according to the manufacturer's instructions.

**Table 2 Hydrogen bonding interactions between MMβA-mono and the Stx2a A-subunit.**

| MMβA-mono | Wat | Stx2 | Hydrogen bond | | Distance |
|-----------|-----|------|---------------|---|----------|
| | | | Donor | Acceptor | |
| Arg6 | Wat101 (G) | Asp94 (A) | Arg6 NH1 | Wat101 | 2.61 |
| Arg6 | Wat101 (G) | Asp94 (A) | Wat101(G) | Asp94 O | 2.93 |
| Arg7 | | Thr115 (A) | Arg7 NH2 | Thr115 OG1 | 3.20 |
| Arg7 | | Thr115 (A) | Arg7 NH1 | Thr115 OG1 | 3.28 |
| Arg7 | | Asp70 (D)* | Arg7 NH2 | Asp70 OXT | 2.76 |
| Arg7 | | Asp70 (D)* | Arg7 NE | Asp70 OXT | 3.02 |
| Arg7 | Wat102 (G) | Lys5 (D)* | Arg7 NH1 | Wat102 O | 2.72 |
| Arg7 | Wat102 (G) | Lys5 (D)* | Wat102 | Lys5 O | 3.27 |
| Arg8 | | Tyr114 (A) | Tyr114 N | Arg8 O | 3.07 |
| Arg8 | Wat104 (G) | Thr115 (A) | Arg8 N | Wat104 | 3.46 |
| Arg8 | Wat104 (G) | Thr115 (A) | Wat104 | Thr115 OG1 | 2.92 |
| Arg8 | | Thr199 (A) | Arg8 NH2 | Thr199 O | 2.91 |
| Arg8 | | Glu167 (A) | Arg8 NH2 | Glu167 OE1 | 3.09 |
| Arg8 | | Glu167 (A) | Arg8 NH1 | Glu167 OE2 | 2.86 |
| Arg8 | Wat105 (G) | Gly203 (A) | Gly203 N | Wat105 O | 3.17 |
| Arg8 | Wat105 (G) | Gly203 (A) | Arg8 NE | Wat105 O | 3.01 |
| Arg9 | | Asp94 (A) | Arg9 NE | Asp94 OD2 | 2.94 |
| Arg9 | | Asp94 (A) | Arg9 NH2 | Asp94 O | 3.04 |
| Arg9 | Wat318 (A) | Tyr77 (A) | Arg9 N | Wat318 O | 2.91 |
| Arg9 | Wat318 (A) | Tyr77 (A) | Wat318 O | Tyr77 OH | 2.57 |
| Arg9 | Wat346 (A) | Glu72 (A) | Arg9 NH2 | Wat346 O | 3.01 |
| Arg9 | Wat346 (A) | Glu72 (A) | Arg9 NH1 | Wat346 O | 3.26 |
| Arg9 | Wat346 (A) | Glu72 (A) | Wat346 O | Glu72 OE1 | 2.70 |
| Ala10 | | Val78 (A) | Val78 N | Ala10 O | 2.83 |
| Ala10 | | Ser112 (A) | Ala10 NT | Ser112 OG | 2.99 |
| Ala10 | | Val78 (A) | Ala10 NT | Val78 O | 3.01 |
| Ala10 | | Ser112 (A) | Ala10 N | Ser112 OG | 3.56 |
| Ala10 | Wat103 (G) | Arg170 (A) | Arg170 NH1 | Wat103 O | 2.90 |
| Ala10 | Wat103 (G) | Arg170 (A) | Arg170 NH2 | Wat103 O | 3.05 |
| Ala10 | Wat103 (G) | Arg170 (A) | Wat103 O | Ala10 O | 2.77 |

The asterisk shows the residue of the Stx2a B-subunit.

**Binding assay between Stx and peptides.** The AlphaScreen assay was used to assess the binding between the Stx A-subunit or B-subunit pentamer and peptides. For the binding between the B-subunit pentamer and peptides, various concentrations of biotinylated peptides were incubated with 1aBH, 2aBH, or its mutant (260 nM) in individual wells of an OptiPlate-384 (PerkinElmer, MA, USA) for 30 min at room temperature. The samples were then incubated with nickel chelate acceptor beads (20 μg/ml; PerkinElmer) for 30 min, followed by incubation with streptavidin donor beads (20 μg/ml; PerkinElmer) for 1 h at room temperature in the dark. The plate was then subjected to excitation at 680 nm, and emission from the wells was monitored at 615 nm with an EnVision system (PerkinElmer). Data were obtained as the arbitrary unit (AU) of signal intensity (counts per second). 1aBH-F30A, -A56Y, and -W34A are site 1-, site 2-, and site 3-mutant, respectively. 2aBH-W29A, -G61A, and -W33A are site 1-, site 2-, and site 3-mutant, respectively. 2aBH-W29A/G61A/W33A is a site 1, 2, and 3 triple mutant. The precise receptor-binding features of these mutants have been confirmed in our previous study[28].

For the binding between the A-subunit and peptides, various concentrations of biotinylated peptides were incubated with the Stx1a A-subunit or Stx2a A-subunit (20 nM) in the presence of each specific anti-Stx A-subunit monoclonal antibody (originally obtained) in individual wells of an OptiPlate-384 (PerkinElmer) for 30 min at room temperature. The samples were then incubated with anti-IgG (protein A) acceptor beads (20 μg/ml; PerkinElmer) for 30 min, followed by incubation with streptavidin donor beads (20 μg/ml; PerkinElmer) for 1 h at room temperature in the dark. Data were obtained as the AU of signal intensity, as described above. To evaluate the importance of each amino acid within MMβA-mono on Stx2a A-subunit binding, a series of MMβA-monomers with amino acid substitution(s) were used.

**ELISA of the binding between the B-subunit pentamer and peptides.** The indicated amounts of biotinylated MMβA-tet or MMβA-mono (10 μg/ml) were applied onto each well of a regular 96-well ELISA plate (Thermo Fisher Scientific, MA, USA) and incubated overnight for 24 h. After blocking, the plate was incubated with various concentrations of 1aBH or 2aBH for 1.5 h at room temperature. Bound 1aBH/2aBH was detected using mouse monoclonal anti-His-tag antibody (clone: 9C11, Wako Pure industries, Japan) and horseradish peroxidase-conjugated, horse anti-mouse IgG antibody (Cell Signaling Technology, MA, USA).

**Crystallization.** Purified Stx2a holotoxin was concentrated to 4–8 mg/ml in 0.2 M NaCl and 25 mM potassium phosphate (pH 6.5), using Amicon Ultra-0.5 (10-kDa cutoff). The crystallization condition was optimized by the hanging-drop vapor diffusion method, based on the previously reported crystallization conditions of Stx2 holotoxin[19]. The best crystallization condition was 4.0 M sodium formate, 100 mM 2-(N-morpholino) ethanesulfonic acid (MES) pH 6.5, 50 mM 3-(1-Pyridinio)-1-propanesulfonate (PPS), and 2% ethylene glycol. The micro-seeding method was utilized to obtain crystals with high reproducibility. To prepare a complex with MMβA-mono, Stx2 holotoxin crystals were soaked in artificial mother liquor [4.0 M sodium formate, 70 mM MES pH 6.5, 35 mM PPS, 1.4% (v/v) ethylene glycol] containing 5 mM MMβA-mono for 1.5 h. The crystals were then cryoprotected in cryoprotectant solution [30% (v/v) ethylene glycol, 2.8 M sodium formate, 70 mM MES pH 6.5, 35 mM PPS] containing 5 mM MMβA-mono for 15 s.

**Diffraction data collection and structure determination.** Diffraction data for apo form and MMβA-mono complex crystals were collected at 95 K on beamlines BL-1A and BL-17A of Photon Factory at KEK (Tsukuba, Japan), respectively. The diffraction data were processed and scaled using the programs *XDS* and *XSCALE*[29]. Both crystals belonged to space group $P6_1$ (Table 1). The crystal structures of Stx2 holotoxin (PDB ID: 7D6Q) and its complex with MMβA-mono (PDB ID: 7D6R) were determined by the molecular replacement (MR) method using the PHENIX program[30]. The PDB coordinates of 1R4P (Shiga toxin type 2)[19] were used as a search model for the MR calculations. Crystallographic refinements were performed using the phenix.refine program[30]. The surface electrostatic potential was calculated in PyMOL (Schrödinger, NY).

**Statistics and reproducibility.** Significant differences between the two groups were analyzed using an unpaired two-sided Student's *t* test. Significant differences between each group and the control group were analyzed using one-way analysis of variance followed by Dunnett's test or Dunnett's T3 test based on the equality of two variances. All statistical analysis was performed using IBM SPSS Statistics software (ver. 27.0.0.0). No statistical methods were used to determine the sample size. We repeated each experiment at least three times and confirmed the reproducibility of each result.

**Reporting summary**. Further information on research design is available in the Nature Research Reporting Summary linked to this article.

## Code availability

All source data presented in the main figures and supplementary figures are available in Supplementary Data 1. The refined X-ray structures are available in PDB (PDB ID: 7D6Q, 7D6R). All other data or sources are available from the corresponding authors on reasonable request.

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

## Acknowledgements

This work was supported by grants from the Japan Society for the Promotion of Science (JSPS) KAKENHI (18K07128), the Research Program on Emerging and Re-emerging Infectious Diseases from the Japan Agency for Medical Research and Development (AMED) (JP18fk0108065), The Naito Foundation, Mishima Kaiun Memorial Foundation, and Platform Project for Supporting Drug Discovery and Life Science Research (Basis for Supporting Innovative Drug Discovery and Life Science Research (BINDS)) from AMED under Grant Number JP19am0101071 (support number 0559).

## Author contributions

M.W.-T. and K.N. performed the biochemical experiments, analyzed and interpreted the data, and wrote the manuscript. M.T., M.S., A.O., A.M., and T.S. crystallized proteins and collected the diffraction data for the crystals. M.S and T.S. performed crystallographic analysis and interpreted the data. M.T. and M.H. performed the biochemical experiments and analyzed the data. M.W.-T. and E.S. synthesized the peptides. K.N., T.S., and A.M. supervised the project.

## Competing interests

The authors declare no competing interests.
