## [Peer Review File · Communications Biology]

Reviewers' Comments:

Reviewer #1:

Remarks to the Author:

1. Brief summary of the manuscript

The manuscript identifies two inhibitors of Stx toxins, one of which binds to the active site of the A-subunit and one of which binds to the B-pentamer. The connection between the two inhibitors is that the one binding to the B-pentamer is a molecule made of four copies of the inhibitor of the A-subunit covalently linked via a scaffolding unit. As well as the inhibition and binding studies, the crystal structure of the holotoxin with the active-site inhibitor is presented, showing the interactions between the holotoxin and the inhibitor.

2. Overall impression of the work

The work is well done with data that support the conclusions. One question I had is the relevance of the relationship between the two inhibitors. How could this information be used to design a better inhibitor, one that would target either the active site or the receptor-binding site?

The role of beta-Ala is not obvious, since clear electron density is only shown for the backbone atoms of residues 6 to 10 (Supplementary Figure 2). Since residue 10 appears to be buried, it is clear why the tetravalent peptide inhibitor would not bind in this site.

I checked that the PDB ID indicated that the crystallographic model had been deposited and was awaiting validation. I did not receive the validation reports, but the table in the supplementary material and the resolution of the data give me no concerns.

3. Specific comments, with recommendations for addressing each comment.

The important comments are 2) and 9), since I was confused about the notation and other readers might also be confused.

1) Line 55 remove "that of"

2) Line 58 The notation in parentheses was not clear to me.

3) Line 68 I would cut "image". A diffraction data set is collected over many images, so the use of the term here is confusing. I would say "Based on the 1.6 Å-resolution diffraction" or "Based on the diffraction to 1.6 Å-resolution".

4) Line 81 interact (since this is ongoing)

5) Lines 111, 134, 136, 156 capitalize "Shiga"

6) Line 172 amino acid (Only one replaces M or A.)

7) Line 193-5 The surface of Stx2A is colored by charge (blue, positive; red, negative).

8) Line 199-200 shown as spheres.

9) Line 265, Figure 1. Would it be possible to split the figure into more parts? Part (a) is the one I struggled to understand, since it shows the structure as well as blots in the panels. I struggled with the top figure, in part because of the choice of R to represent the group that contains R, the arginine residue. Arrows are usually used to represent beta-strands and the connection between the turquoise dashed rectangle and the turquoise ribbons was not clear to me. Not being a scientist doing peptide synthesis, I don't know what the grey bar at the bottom of the picture under the three NHs represents.

I would advise using B-pentamer instead of B-subunit whenever it is clear that the B-subunits must be assembling into the pentamer. In the figure, why is BH used?

Reviewer #2:

Remarks to the Author:

Watanabe-Takahashi et al. identified the peptide motifs showing inhibitory effect on the

cytotoxicity of Shiga toxin, Stx, such as tetrameric form of Met-Met-betaAla-Arg-Arg-Arg-Arg, referred to as MMbA-tet. Its monomeric form, MMbA-mono, exhibits lower cytotoxicity than MMbA-tet. Furthermore, the binding study using the Alpha Screen revealed that the MMbA-tet binds to B subunit, while MMbA-mono to A subunit. In addition, they determined the crystal structure of the Stx complexed with its monomeric form, MMbA-mono. The structure shows that the MMbA-mono mainly binds to A subunit of Stx and one Arg residue interacts with B subunit. This paper is somehow interesting and has potential to advance our knowledge for a field of toxin inhibitors. I have following comments.

Comments

1. The whole main text seems too short to be easily understood because essential description, such as experimental methods and discussion, is not enough.
2. The first appearance of "MMA-tet", "MMbA-tet" or "MMbA-mono" in the text should be clearly defined.
3. The difference of inhibitory effects, IC₅₀, between MMbA-tet and MMbA-mono is not huge. The authors should discuss the reason.
4. The crystal structure only includes the betaAla-Arg-Arg-Arg-Arg part but not the N-terminal Met-Met part. The authors should discuss whether the N-terminal site of MMbA-tet has some effect on the tetramerization and Stx binding.
5. The authors should discuss why MMbA-tet cannot bind to A subunit but B subunit, while MMbA-mono can bind to A subunit. Since the tetramerization of the Arg-Arg-Arg-Arg part may possibly induce less specific binding, the authors should show how much specific binding of MMbA-tet to Stxs.
6. Figure 1b and d. Cell viability for Stx2a without any peptide is 40% (that for Stx1a is 20%, which still seems high?). The concentration of Stx2a may be too small for appropriate inhibition assays to determine the IC₅₀ value.
7. Figure 1e. In the ELISA experiment, the streptavidin-immobilized plate was used?
8. Figure 1e. Why did not the authors perform the AlphaScreen for the binding to B subunit?
9. Figure 1e. Stx1a binding to MMbA-tet shows 40% binding at 0.1 uM, and missing at less than 0.1 uM, where MMbA-mono binding was performed. Why did not the authors perform the binding experiments at lower concentrations?

Response to comments from Reviewer #1:

We would like to thank Reviewer #1 for considering our manuscript and offering constructive comments that help strengthen our conclusions. We have provided point-by-point responses to the Reviewer's concerns below.

Reviewer #1 (Remarks to the Author):

1. Brief summary of the manuscript

The manuscript identifies two inhibitors of Stx toxins, one of which binds to the active site of the A-subunit and one of which binds to the B-pentamer. The connection between the two inhibitors is that the one binding to the B-pentamer is a molecule made of four copies of the inhibitor of the A-subunit covalently linked via a scaffolding unit. As well as the inhibition and binding studies, the crystal structure of the holotoxin with the active-site inhibitor is presented, showing the interactions between the holotoxin and the inhibitor.

2. Overall impression of the work

The work is well done with data that support the conclusions. One question I had is the relevance of the relationship between the two inhibitors. How could this information be used to design a better inhibitor, one that would target either the active site or the receptor-binding site?

Response: As the Reviewer points out, it will be important to use our data to design a better inhibitor. One possibility is to design an inhibitor that could target both the catalytic A-subunit and the receptor binding B-subunit simultaneously by using the common essential binding motif, RRRRA or RRRR. For this purpose, it is feasible to assemble the motifs in tandem in a monomeric peptide with the C-terminus, RRRRA or RRRR, in which the C-terminus motif can bind to the catalytic A-subunit, and then the clustered motifs can bind to the B-subunit pentamer based on the clustering effect. We have already found that a receptor mimic-compound with a linear structure containing only two trisaccharide moiety of Gb3 (Gal α [1-4]-Gal β [1-4]-Glc β -) at each end, which is referred to as SUPER TWIG(1)2, can exert the clustering effect sufficiently to bind to the B-subunits of Stx1a and Stx2a with K_d values of 88 μ M and 68 μ M, respectively, when compared to the K_d value of the free trisaccharide (mM order) (K. Nishikawa et. al., J. Infect. Dis., 2005, 191, 2097-). This suggests that we can increase the binding affinity to the B-subunit by tandemly clustering the binding motifs even for a monomeric peptide. In our preliminary data, we found that a highly positively charged protein, protamine, which is enriched in salmon sperm and has the following amino acid sequence, MPRRRRASRRVRRRRRPRVSRRRRRGRRRR, and its trypsin-digested fragments inhibited the

cytotoxicity of Stx2a more effectively than MM β A-mono, supporting the possible feasibility of such a novel type of Stx inhibitor.

Another approach is to identify a small molecule that binds to and inhibits Stx by screening a synthetic compound library based on a pharmacophore obtained from our present data. Because this project is ongoing, we mentioned this approach in the revised version of the manuscript (lines 184-186).

The role of beta-Ala is not obvious, since clear electron density is only shown for the backbone atoms of residues 6 to 10 (Supplementary Figure 2). Since residue 10 appears to be buried, it is clear why the tetravalent peptide inhibitor would not bind in this site.

Response: As the reviewer mentions, it is likely that β Ala is required for the high-affinity binding of MM β A-tet with the B-pentamer, but not for the binding of MM β A-mono with the A-subunit. Fig 1c in the revised version of the manuscript clearly indicates the importance of β Ala in the inhibitory effect of MM β A-tet, while our recent data showed that a series of shorter peptides, such as ARRRRA, RRRRA, or RRRR, still efficiently inhibited the binding between MM β A-mono and the A-subunit (unpublished data), supporting the idea that β Ala is not required for the binding of MM β A-mono with the A-subunit. We mentioned this in the revised version of the manuscript (lines 191-195).

I checked that the PDB ID indicated that the crystallographic model had been deposited and was awaiting validation. I did not receive the validation reports, but the table in the supplementary material and the resolution of the data give me no concerns.

Response: We sincerely apologize for the inconvenience. We re-registered the crystallographic data and attached the new validation reports with the revised version of the manuscript, and the new PDB IDs are indicated in Table 1.

3. Specific comments, with recommendations for addressing each comment.

The important comments are 2) and 9), since I was confused about the notation and other readers might also be confused.

1) Line 55 remove “that of”

Response: Following the reviewer’s suggestion, we changed the sentences as follows;

“Among these peptides, MMβA-tet, which has a synthetic amino acid βAla in its motif, better inhibited the cytotoxicity of Stx1a and Stx2a, with relative IC50 values (concentration that restored viability to 50% of the cells that were killed when no peptide was added) of 1.9 μM and 2.7 μM, respectively (Fig. 1c).”

in the revised version of the manuscript (lines 94-98).

2) Line 58 The notation in parentheses was not clear to me.

Response: Following the reviewer’s comment, we changed the sentence as follows;

“Unexpectedly, a monomeric peptide with the same motif,

Met-Ala-Met-Met-βAla-Arg-Arg-Arg-Arg-Ala (referred to as MMβA-mono), also inhibited the cytotoxicity of Stx1a and Stx2a with greater potency against Stx2a (relative IC50 = 59 and 9.6 μM, respectively) (Fig. 2a).”

in the revised version of the manuscript (lines 110-113).

3) Line 68 I would cut “image”. A diffraction data set is collected over many images, so the use of the term here is confusing. I would say “Based on the 1.6 Å-resolution diffraction” or “Based on the diffraction to 1.6 Å-resolution”.

Response: Following the reviewer’s suggestion, we changed the sentence as follows;

“Diffraction to 1.6 Å-resolution showed that MMβA-mono was bound tightly to the catalytic A-subunit through electrostatic interactions of the Arg-cluster with Asp94 and Glu167 of the A-subunit and with Asp70 of the B-subunit, residues that are all present in the catalytic cavity (Figs. 3a, 3b, and Table 2).”

in the revised version of the manuscript (lines 126-129).

4) Line 81 interact (since this is ongoing)

Response: Following the reviewer’s suggestion, we changed the sentence as follows;

“Also, Glu167 and Arg170, both of which are known to be essential for catalytic activity, were found to interact with Arg at position 8 of MMβA-mono.”

in the revised version of the manuscript (lines 142-143).

5) Lines 111, 134, 136, 156 capitalize “Shiga”

Response: Following the reviewer's suggestion, we changed the word "shiga" to "Shiga" in the revised version of the manuscript (lines 365, 396, 406, 409).

6) Line 172 amino acid (Only one replaces M or A.)

Response: Following the reviewer's suggestion, we changed the word "amino acids" to "amino acid" in the revised version of the manuscript (line 461).

7) Line 193-5 The surface of Stx2A is colored by charge (blue, positive; red, negative).

Response: Following the reviewer's suggestion, we changed the following sentence, "The surface is colored according to the electrostatic potential of the residues (blue, positive; red, negative)."

to

"The surface of the Stx2a A-subunit is colored by charge (blue, positive; red, negative)." in the revised version of the manuscript (lines 486-487).

8) Line 199-200 shown as spheres.

Response: Following the reviewer's suggestion, we changed the words "sphere models" to "spheres" in the revised version of the manuscript (line 491).

9) Line 265, Figure 1. Would it be possible to split the figure into more parts? Part (a) is the one I struggled to understand, since it shows the structure as well as blots in the panels. I struggled with the top figure, in part because of the choice of R to represent the group that contains R, the arginine residue. Arrows are usually used to represent beta-strands and the connection between the turquoise dashed rectangle and the turquoise ribbons was not clear to me. Not being a scientist doing peptide synthesis, I don't know what the grey bar at the bottom of the picture under the three NHs represents.

Response: Following the reviewer's suggestion, we split Fig. 1 into two parts, Figs. 1 and 2 in the revised version of the manuscript. In new Fig. 1a, the structure of the tetravalent peptides synthesized on a cellulose membrane is shown. The gray disc represents the area on the membrane where the tetravalent peptides are synthesized. In the structure, we changed R to Z to represent the motif with Xs. In the new Fig. 1b, the structure of the tetravalent peptides with identified motifs is shown. Accordingly, we changed the legend of Fig. 1 in the revised version of the manuscript (line 464).

I would advise using B-pentamer instead of B-subunit whenever it is clear that the B-subunits must be assembling into the pentamer. In the figure, why is BH used?

Response: Following the reviewer's advice, in the revised version of the manuscript we used "B-subunit pentamer" instead of "B-subunit" when it is clear that the B-subunits must be assembling into the pentamer (lines, 31, 206, 208, 252, 277, 455, 480).

BH indicates that we used histidine-tagged B-pentamers in this study. Previously, we have shown that the histidine-tagged Stx1a B- and Stx2a B-pentamers bind to the trisaccharide moiety of Gb3 based on the clustering effect with similar efficiency to that of holo Stxs (K. Nishikawa et. al., J. Infect. Dis., 2005, 191, 2097-). Furthermore, we have already determined the precise receptor-binding features of a series of the Stx B-pentamer mutants used in this study, all of which are also histidine-tagged. In all cases, the histidine-tag is present on the side of the B-pentamer opposite to the receptor-binding region, having no effect on the receptor binding. We mentioned this in the revised version of the manuscript (lines 263-264).

Additional corrections:

- 1) In Fig. 1b in the original version of the manuscript, we misplaced the left panel for the right panel. We sincerely apologize for the mistake. We corrected this as Fig. 1c in the revised version of the manuscript.
- 2) In the lower panels of Fig. 1e in the original version of the manuscript, we showed the concentration of 1aBH or 2aBH as (μM), which should be ($\mu\text{g/ml}$). We sincerely apologize for the mistake. We replaced the figure as Fig. 2c in the revised version of the manuscript.

Response to comments from Reviewer #2:

We would like to thank Reviewer #2 for reviewing our manuscript and offering constructive comments that help to strengthen our conclusions. We have provided point-by-point responses to the Reviewer's concerns below.

Reviewer #2 (Remarks to the Author):

Watanabe-Takahashi et al. identified the peptide motifs showing inhibitory effect on the cytotoxicity of Shiga toxin, Stx, such as tetrameric form of Met-Met-betaAla-Arg-Arg-Arg-Arg, referred to as

MMbA-tet. Its monomeric form, MMbA-mono, exhibits lower cytotoxicity than MMbA-tet. Furthermore, the binding study using the Alpha Screen revealed that the MMbA-tet binds to B subunit, while MMbA-mono to A subunit. In addition, they determined the crystal structure of the Stx complexed with its monomeric form, MMbA-mono. The structure shows that the MMbA-mono mainly binds to A subunit of Stx and one Arg residue interacts with B subunit. This paper is somehow interesting and has potential to advance our knowledge for a field of toxin inhibitors. I have following comments.

Comments

1. The whole main text seems too short to be easily understood because essential description, such as experimental methods and discussion, is not enough.

Response: Following the reviewer's comment, we have reformatted the manuscript based on the style and formatting guidelines of *Communications Biology*. Accordingly, we have provided more descriptions about the experimental methods and more discussion for a better understanding by the reader.

2. The first appearance of "MMA-tet", "MMbA-tet" or "MMbA-mono" in the text should be clearly defined.

Response: Following the reviewer's suggestion, we have clearly defined "MMA-tet", "MMβA-tet", and "MMβA-mono" at their first appearance in the revised version of the manuscript (lines 62-63, 94-95, and 110-111).

3. The difference of inhibitory effects, IC₅₀, between MMbA-tet and MMbA-mono is not huge. The authors should discuss the reason.

Response: As shown in Fig. 2a in the revised version of the manuscript, each apparent K_d value of MMβA-mono for the A-subunit of Stx1a or Stx2a (0.032 or 0.05 μM, respectively) is similar to that of MMβA-tet for the binding to the B-subunit pentamer of Stx1a or Stx2a (0.03 or 0.07 μM, respectively), indicating that the binding affinity of MMβA-mono for the A-subunit is similar to that of MMβA-tet for the B-subunit pentamer. In contrast, in the cytotoxicity assay, the relative IC₅₀ value of MMβA-mono for Stx1a or Stx2a (59.2 or 9.6 μM, respectively) is higher than that of MMβA-tet (1.9 or 2.7 μM, respectively), indicating that the inhibitory effect of MMβA-mono is lower than that of MMβA-tet, as the reviewer mentions. One possible reason is that the low cell permeability of MMβA-mono might substantially affect its inhibitory efficacy because the target of

the A-subunit is intracellular 28S ribosomal RNA present in the cytosol. Following the reviewer's comment, we included this information in the revised version of the manuscript (lines 163-173).

4. The crystal structure only includes the betaAla-Arg-Arg-Arg-Arg part but not the N-terminal Met-Met part. The authors should discuss whether the N-terminal site of MMbA-tet has some effect on the tetramerization and Stx binding.

Response: The crystal structure includes the Arg-Arg-Arg-Arg-Ala region, but not β Ala-Arg-Arg-Arg-Arg region. As the reviewer points out, we could not see the crystal structure of the Met-Met- β Ala- region. It is likely that Met-Met- β Ala- region is required for the high-affinity binding of MM β A-tet with the B-pentamer, as mentioned below, but not for the binding of MM β A-mono with the A-subunit. Our recent data show that a series of shorter peptides, such as ARRRRA, RRRRA, or RRRR still efficiently inhibited the binding between MM β A-mono and the A-subunit (unpublished data), supporting the above hypothesis. We included this information in the revised version of the manuscript (lines 191-195).

In our previous study, we found that two other tetravalent peptides, AAA-tet or MAA-tet, which have the following motifs, Ala-Ala-Ala-Arg-Arg-Arg-Arg or Met-Ala-Ala-Arg-Arg-Arg-Arg, respectively, did not efficiently inhibit the cytotoxicity of Stx1a and Stx2a compared to MMA-tet (Infect. Immun., 2013, 81, 2133-), indicating that the N-terminal region of MM β A-tet, especially the 2nd Met, plays an essential role in its inhibitory activity against Stx1a and Stx2a. Following the reviewer's comment, we included this information in the revised version of the manuscript (lines 80-87). Since the tetramerization of the peptide is chemically performed using the C-terminal region of the motif to bind to the core structure, as shown in Fig. 1(b) in the revised version of the manuscript, the N-terminal region is not involved in tetramerization.

5. The authors should discuss why MMbA-tet cannot bind to A subunit but B subunit, while MMbA-mono can bind to A subunit. Since the tetramerization of the Arg-Arg-Arg-Arg part may possibly induce less specific binding, the authors should show how much specific binding of MMbA-tet to Stxs.

Response: As shown in Fig. 3 in the revised version of the manuscript, in the complex of MM β A-mono and Stx2a, the C-terminal Ala of MM β A-mono (Met-Ala-Met-Met- β Ala-Arg-Arg-Arg-Arg-Ala) is present in the bottom of the catalytic pocket of the A-subunit. In contrast, in the structure of MM β A-tet, each C-terminal Ala of the motif (Met-Ala-Met-Met- β Ala-Arg-Arg-Arg-Arg-Ala-) is connected to the core structure through a spacer, aminohexanoic acid (U), as shown in Fig 1(b) in the revised version of the manuscript, hampering

the binding of MM β A-tet to the A-subunit through the C-terminal Ala. Following the reviewer's suggestion, we included this information in the revised version of the manuscript (lines 133-136).

In our previous study using PPP-tet, another tetravalent inhibitory peptide against Stx2a with the following motif, Pro-Pro-Pro-Arg-Arg-Arg-Arg, molecular dynamics simulations demonstrated the interaction of each Arg with specific acidic amino acids present in the receptor-binding region of the B-pentamer, which was confirmed by biochemical binding analysis using a series of the B-pentamer mutants with amino acid substitutions at the acidic amino acids (K. Nishikawa et. al., FASEB J., 2006, 20, 2597-). Based on these findings, we performed a binding assay of MM β A-tet to the B-pentamer using the B-pentamer mutants and found that MM β A-tet specifically bound to the B-pentamers of Stx1a and Stx2a through site 3 and sites 1-3, respectively (as shown in Fig. 1d), where the acidic amino acids are substantially involved in the binding. Following the reviewer's suggestion, we included this information in the revised version of the manuscript (lines 81-82, 99-107, 195-200).

6. Figure 1b and d. Cell viability for Stx2a without any peptide is 40% (that for Stx1a is 20%, which still seems high?). The concentration of Stx2a may be too small for appropriate inhibition assays to determine the IC₅₀ value.

Response: In Fig. 1b in the original manuscript, we misplaced the left panel for the right panel. We sincerely apologize for the mistake. We corrected this as Fig. 1c in the revised version of the manuscript. In order to see the efficient inhibitory effects of the tetravalent peptides, such as PPP-tet and MMA-tet, on the cytotoxicity of Stxs, we have used conditions in which the cell viability for Stx without any peptides is about 20-40% (FASEB J., 2006, 20, 2597-, Infect. Immun., 2013, 81, 2133-, Infect. Immun., 2016, 84, 2653-). For in vivo experiments, however, both PPP-tet and MMA-tet show potent inhibitory effects against the lethality in EHEC infected mice (FASEB J., 2006, 20, 2597-, Infect. Immun., 2013, 81). Similarly, under the condition in which the cell viability for Stx without MM β A-mono was lower than 20%, we did not see the efficient inhibitory effects of MM β A-mono. Therefore, we used the conditions described in this study. To address the reviewer's concern, in the revised version of the manuscript we re-calculated the relative IC₅₀ values to appropriately evaluate the inhibitory effects of the peptides (lines 96-98, 112-113); the relative IC₅₀ value was determined as the concentration of each peptide that restored viability to 50% of the cells that were killed when no peptide was added.

7. Figure 1e. In the ELISA experiment, the streptavidin-immobilized plate was used?

Response: In this ELISA experiment, we used a regular ELISA plate (Thermo Fisher Scientific), not a streptavidin-immobilized one, to immobilize biotinylated MM β A-mono. In this case, the

immobilization efficacy of biotinylated MM β A-mono was much better than that of free MM β A-mono. Following the reviewer's comment, we added the information about the ELISA plate in the revised version of the manuscript (line 279).

8. Figure 1e. Why did not the authors perform the AlphaScreen for the binding to B subunit?

Response: Following the reviewer's suggestion, we performed the AlphaScreen assay for the binding of the peptides to the B-pentamers. MM β A-tet efficiently bound to both Stx1a B- and Stx2a B-pentamers, consistent with the data shown in Fig. 1d and Fig. 2c in the revised version of the manuscript. In contrast, the binding of MM β A-mono to the B-pentamers was markedly lower, consistent with the data shown in Fig. 2c in the revised version of the manuscript. We included the new data in the revised version of the manuscript (Fig. 2b, lower panels, lines 113-119).

9. Figure 1e. Stx1a binding to MMbA-tet shows 40% binding at 0.1 uM, and missing at less than 0.1 uM, where MMbA-mono binding was performed. Why did not the authors perform the binding experiments at lower concentrations?

Response: Following the reviewer's suggestion, we performed the AlphaScreen assay for the binding to the A-subunits with lower concentrations of the peptides. We could clearly show that MM β A-tet does not bind to either A-subunit at all. We showed these data in the revised version of the manuscript (Fig. 2b upper panels, lines 115-119).

Additional correction: In the lower panels of Fig. 1e in the original version of the manuscript, we showed the concentration of 1aBH or 2aBH as (μ M), which should be (μ g/ml). We sincerely apologize for the mistake. We replaced the figure as Fig. 2c in the revised version of the manuscript.

Reviewers' Comments:

Reviewer #1:

Remarks to the Author:

The authors have done a good job of responding to comments by the other reviewer and myself. Since there were so many changes to the writing, I have a few editorial comments, but think the revised version is acceptable. Here is a list of my editorial comments:

Line 34 replace including with interacting with

Line 36 I am hesitating on functionally independent, wondering if this should be functionally distinct. Although the subunits have different functions (B for binding, A for catalysis), these functions are not independent since delivery of A requires B. This expression is in the title of the paper as well. (also line 201)

Line 50 communis

Line 51 similar instead of homologous. Proteins are homologous or not, they can't be more or less homologous

Line 57 sites

Line 66 and 68 move "with greater potency than MMA-tet" after tetravalent peptide

Line 80 2nd to second

Line 89 "whose structure was optimized for Stx" is not clear. "Whose" cannot be referring to the cellulose membrane. If this is a screening experiment, how can the structure be optimized? Please clarify.

Line 90 add "and" before the radioactivity

Lines 100 and 115 add "the" between with and AlphaScreen

Line 148 I suggest changing "consistent with the" to suggesting a

Line 456 What is the antecedent for "which"? If it is peptides, "which were" should be used.

Reviewer #2:

None

Response to comments from Reviewer #1:

We would like to thank Reviewer #1 for considering our manuscript and offering constructive comments. We have provided point-by-point responses to the Reviewer's concerns below.

Reviewer #1 (Remarks to the Author):

The authors have done a good job of responding to comments by the other reviewer and myself. Since there were so many changes to the writing, I have a few editorial comments, but think the revised version is acceptable. Here is a list of my editorial comments:

Line 34 replace including with interacting with

Response: Following the reviewer's suggestion, we changed the text (line 34) in the resubmitted version of the manuscript.

Line 36 I am hesitating on functionally independent, wondering if this should be functionally distinct. Although the subunits have different functions (B for binding, A for catalysis), these functions are not independent since delivery of A requires B. This expression is in the title of the paper as well. (also line 201)

Response: Following the reviewer's suggestion, we changed the expression "independent" to "distinct" in the title and the text (lines 2, 36, 203) in the resubmitted version of the manuscript.

Line 50 communis

Response: Following the reviewer's suggestion, we changed the text (line 50) in the resubmitted version of the manuscript.

Line 51 similar instead of homologous. Proteins are homologous or not, they can't be more or less homologous

Response: Following the reviewer's suggestion, we changed the text (line 51) in the resubmitted version of the manuscript.

Line 57 sites

Response: Following the reviewer's suggestion, we changed the text (line 57) in the resubmitted version of the manuscript.

Line 66 and 68 move "with greater potency than MMA-tet" after tetravalent peptide

Response: Following the reviewer's suggestion, we changed the text (lines 66-68) in the resubmitted version of the manuscript.

Line 80 2nd to second

Response: Following the reviewer's suggestion, we changed the text (line 80) in the resubmitted version of the manuscript.

Line 89 "whose structure was optimized for Stx" is not clear. "Whose" cannot be referring to the cellulose membrane. If this is a screening experiment, how can the structure be optimized? Please clarify.

Response: Following the reviewer's suggestion, we changed the following text,

“To find peptides with stronger inhibitory activity than MMA-tet, we screened a series of tetravalent peptides with randomized amino acids synthesized on a cellulose membrane whose structure was optimized for Stx (Fig. 1a). “

to the text as follows:

“To find peptides with stronger inhibitory activity than MMA-tet, we screened a series of tetravalent peptides with randomized amino acids synthesized on a cellulose membrane (Fig. 1a). The density of tetravalent peptides on the membrane and the spacer length were optimized for the screening using Stx as shown previously^{23,24}.” (lines 87-91) in the resubmitted version of the manuscript.

Line 90 add "and" before the radioactivity

Response: Following the reviewer's suggestion, we changed the text (line 91) in the resubmitted version of the manuscript.

Lines 100 and 115 add "the" between with and AlphaScreen

Response: Following the reviewer's suggestion, we changed the text (lines 102, 117) in the resubmitted version of the manuscript.

Line 148 I suggest changing "consistent with the" to suggesting a

Response: Following the reviewer's suggestion, we changed the text (line 150) in the resubmitted version of the manuscript.

Line 456 What is the antecedent for "which"? If it is peptides, "which were" should be used.

Response: Following the reviewer's suggestion, we changed the following text,

“Structure of the tetravalent peptides synthesized on a cellulose membrane, which was optimized to Stx. The density of the tetravalent peptide was set to 100% by using”

to the text as follows:

“Structure of the tetravalent peptides synthesized on a cellulose membrane is shown. The density of tetravalent peptides on the membrane and the spacer length were optimized for the screening using Stx as follows. The density was set to 100% by using” (lines 469-471) in the resubmitted version of the manuscript.